# Unsupervised Graph Neural Networks for Solving Combinatorial Optimization Problems by Iterative Solution Refinement

## Abstract

Combinatorial optimization (CO) problems are crucial in various scientific and industrial applications. Graph Neural Networks (GNNs) have recently emerged as a scalable, high-performance framework for tackling NP-hard CO problems, demonstrating high performance and nearly linear scalability. Current approaches utilize GNNs to directly predict solutions based on standard node features. This often leads to overfitting and convergence to poor local minima, limiting solution quality. We introduce a novel optimization method leveraging the power of GNNs to efficiently process CO problems with Quadratic Unconstrained Binary Optimization (QUBO) formulation. Rather than predicting a solution from fixed features, our model iteratively refines its output by feeding predictions from each step back as dynamic node inputs. We further enhance performance by modifying the GNN architecture and incorporating informative static features. We evaluate our approach on canonical CO benchmarks including Max-Cut, Graph Coloring, and Maximum Independent Set. Our method significantly outperforms prior learning-based approaches and matches state-of-the-art heuristics, while scaling more efficiently to large instances.

## 1 Introduction

Combinatorial Optimization (CO) is a well-known subject in computer science, bridging operations research, discrete mathematics and optimization. Informally, given some ground set, the CO problem is to select the combination of its elements, such that it lies on the problem's feasible domain and the cost of this combination is minimized. A significant amount of CO problems are known to be NP-hard, meaning that they are computationally intractable under "$P \neq NP$" conjecture and the scope of application for exact algorithms to solve them is very narrow. Therefore, the development of heuristic methods that provide high-accuracy solutions in acceptable amount of time is a crucial challenge in the field Boussaïd et al. (2013).

Many CO problems naturally arise on graph-structured data, where solutions correspond to selected subsets of nodes or edges. Implicit regularities and patterns often arise in graph structure and features, making the use of machine learning and especially graph neural networks (GNNs) very promising Cappart et al. (2023). Researchers from Amazon Schuetz et al. (2022a;b); Wang et al. (2023) propose to apply GNN for solving CO problems with Quadratic Unconstrained Binary Optimization (QUBO) formulation in unsupervised manner. Their approach (PI-GNN) minimizes a differentiable continuous relaxation of the QUBO objective, enabling gradient-based training without labels. Since no training data is required, the GNN is trained end-to-end on each problem instance, effectively acting as an autonomous learned heuristic. Ichikawa (2024) proposed the annealing penalty term improving the PI-GNN performance. A key advantage of this framework is its ability to handle massive graphs with millions of nodes while outperforming traditional heuristics in computational efficiency. However, its effectiveness in producing high-quality solutions has been questioned, with studies showing that even simple greedy algorithms can outperform it Boettcher (2023); Angelini and Ricci-Tersenghi (2023). In addition, existing analyses indicate that GNNs have a tendency to get stuck in local optima when trained for particular problem instances Wang and Li (2023).

To address these limitations, we propose a novel GNN-based framework for solving CO problems formulated as QUBO, leveraging the strengths of unsupervised training. In previous approaches, training is based on a scenario where GNNs have to predict a solution to an input set of properties within a single step. In contrast, we introduce an iterative refinement mechanism, where the GNN incrementally improves the solution until a termination criterion is met. At each step, predictions from the previous iteration are fed back as dynamic node features. Experiments on the canonical benchmark datasets show the advantage of this approach for GNNs of different types. To further improve the quality, we optimized the GNN architecture and selected a set of additional vertex properties. Thus, the main contribution of our work are as follows:

- We propose a new GNN-based framework for solving QUBO-formulated CO problems via iterative refinement, where the GNN receives signal from previous steps to improve the current solution.

- We show that this design significantly improves performance across various GNN types.

- We study the effect of architectural choices and static node features on solution quality and identify an optimal configuration—referred to as **QIGNN** (QUBO-based Iterative GNN) that outperforms state-of-the-art learning-based methods on Max-Cut, Graph Coloring, and Maximum Independent Set. QIGNN also matches or surpasses classical heuristics in solution quality, while offering superior scalability on large graphs.

## 2 PRELIMINARIES

### 2.1 QUBO PROBLEM

One notable CO problem is the Quadratic Unconstrained Binary Optimization (QUBO), which seeks to minimize a pseudo-Boolean polynomial $\mathcal{F}(x)$ of degree two Hammer and Rudeanu (1969); Boros and Hammer (1991):

$$\min_{x \in \{0,1\}^n} \mathcal{F}(x) = \sum_{i=1}^{n} \sum_{j=1}^{n} A_{ij} x_i x_j + \sum_{i=1}^{n} c_i x_i = x^T Q x, \tag{1}$$

where $A \in \mathbb{R}^{n \times n}$ is a symmetric matrix of coefficients, and $c \in \mathbb{R}^n$ represents the linear terms of the objective function. The vector $x = (x_1, x_2, \ldots, x_n)^T$ denotes the binary variables, with each $x_i \in \{0, 1\}$. The matrix $Q \in \mathbb{R}^{n \times n}$ is defined as $Q = A + \text{diag}(c)$, where $\text{diag}(c)$ represents the diagonal matrix with entries from the vector $c$. The equivalence in Equation 1 holds because $x_i^2 = x_i$ for all $i$, which allows the quadratic form to be expressed in terms of the matrix $Q$.

Despite the QUBO problem has been studied for a very long time Hammer and Rubin (1970), it has recently attracted much attention as a way to formulate other CO problems Glover et al. (2022); Lucas (2014), mainly due to the emerging interest in the development of quantum computational devices Boixo et al. (2013); Wang et al. (2013). By applying simple reformulation techniques, such as constraint penalization Smith et al. (1997), a huge number of CO problems can be formulated as QUBO, which makes algorithms for its solution especially valuable in practice.

### 2.2 GRAPH NEURAL NETWORKS FOR COMBINATORIAL OPTIMIZATION

Graph neural networks are capable to learn complex graph-structured data by capturing relational information. During training process, each of the nodes is associated with a vector which is updated based on the information from neighboring nodes.

Let us consider an undirected graph $G = (V, E)$ with a vertex set $V = \{1, \ldots, n\}$ and an edge set $E = \{(i, j) : i, j \in V\}$. Let $h_i^l \in \mathbb{R}^{m_l}$ be a feature vector for a node $i$ and $h_j^l \in \mathbb{R}^{m_l}$ a vector for a node $j$ at the $l$-th convolution layer and let $e = (i, j)$ be an edge between nodes $i$ and $j$.

In this paper, we consider GNNs that are based on the message passing protocol to exchange information between nodes. This protocol consists of the two main parts: message accumulation and message aggregation. Message accumulation computes a message $m_e^{l+1}$ for an edge $e$ using a function $\phi$, which determines how information will be collected. Message passing includes aggregation of collected messages from a node $i$ neighbors by function $\rho$ and then an update of a feature vector $h_i^l$

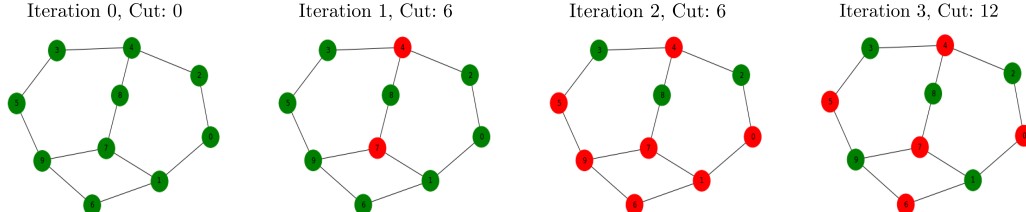

Figure 1: The example of algorithm work on a toy graph of 10 vertices and 12 edges on the Max-Cut problem. The Max-Cut involves partitioning of graph's nodes into two sets such that the number of edges between sets is maximized. At each iteration, the rounded (discretized) solution is shown: red color refers to $x_i = 0$, green to $x_i = 1$. Using intermediate states, the algorithm aims to reclassify each node to the opposite class of the majority of its nearest neighbors at each iteration, thereby optimizing the cut.

for the node $i$ by applying function $f$ with trainable weights $W^l$. So the whole process is described as follows:

$$
\begin{aligned}
m_e^{l+1} &= \phi\left(h_i^l, h_j^l\right), \\
h_i^{l+1} &= f\left(h_i^l, \rho\left(\{m_e^{l+1} : (i,j) \in E\}\right)\right).
\end{aligned}
\tag{2}
$$

There are a variety of approaches how to define input features $h_i^0$ for the node $i$. It could consist of a one-hot encoding vector of a node label, a random or shared dummy vector Cui et al. (2022), a pagerank Brin and Page (1998) or a degree of a node. It is also possible to use a trainable embedding layer before graph convolutions Schuetz et al. (2022a).

To solve a particular CO problem, Schuetz et al. (2022a) proposed to use the continuous relaxation of the QUBO formulation as a loss function for GNNs, introducing their physics-inspired GNN (PI-GNN). Let $\Phi_\theta : G \mapsto [0,1]^n$ denote a GNN whose parameters $\theta$ are shared across all nodes, i.e., parameters associated with $\phi$, $f$ and $\rho$. In our case the same function $f$ is used across layers, but the weights are layer-specific. After the final $N_l$-th convolution the network applies a sigmoid/softmax, yielding the *relaxed* probability vector $p_\theta(\Phi) = \left(p_1(\theta), \ldots, p_n(\theta)\right) \in [0,1]^n$. We treat $p_\theta$ as a continuous surrogate for the binary decision vector $x \in \{0,1\}^n$. Substituting it into the QUBO objective $\mathcal{F}(x) = x^\top Q x$ therefore gives the differentiable *instance-wise* loss function:

$$
\mathcal{L}(\theta; \Phi) = p_\theta(\Phi)^\top Q \, p_\theta(\Phi).
\tag{3}
$$

Learning proceeds by gradient descent on equation 3, without any ground-truth labels. After the final $N_l$-th convolution layer a softmax or sigmoid activation function is applied to compress the final vector $h^{N_l}$ into probabilities $p_\theta$. As a result of training, GNN obtains the continuous solution $p_i(\theta)$ for each node. In order to obtain a solution of the original discrete problem, $p_i(\theta)$ has to be converted into the discrete variable $x_i$. The simplest approach is to apply an indicator function $\mathbb{I}_{p_i > p^*}$ with a threshold $p^*$, as we use in our setup. Alternatively, sampling discrete variables from Bernouli distribution, or greedy methods can be employed Wang et al. (2022).

## 3 METHOD AND TRAINING DESIGN

This section introduces our iterative refinement framework for unsupervised GNN-based combinatorial optimization. We also provide a detailed description of the developed GNN architecture and a selected set of static features.

### 3.1 ITERATIVE FRAMEWORK

**Motivation.** Existing GNNs often rely solely on the static node features, which can lead to significant limitations. When applying GNNs to solve a specific problem instance, this standard approach can

cause the model to become biased towards these features and face node ambiguity issues, resulting in unsupervised GNNs frequently getting trapped in local extrema during training Wang and Li (2023).

Another intuition comes from the structure of the QUBO objective equation 3, which sums interactions over all pairs of active variables. As a result, the optimal decision for a node $i$ i.e., whether $x_i = 0$ or $x_i = 1$ depends strongly on the current states of its neighbors. This suggests that solution quality can be improved by allowing each node to adapt based on the evolving state of the graph.

Based on these two observations, we propose an iterative GNN framework that augments static features with dynamic feedback: each node receives its own predicted probability(state) from the previous iteration as an additional feature. This enables the model to reason over the current solution state via message passing and refine its predictions accordingly. The process defines an iterative optimization loop, where the GNN progressively minimizes equation 3 by adjusting node assignments in response to neighbor updates.

This process is illustrated in Figure 1, which shows the step-by-step work of the algorithm on a toy instance for the maximum cut optimization problem. The algorithm iteratively updates each node's classification, assigning it to the opposite group of the majority of its neighboring nodes, effectively improving the quality of the cut with each step. For this graph, the algorithm is able to achieve the optimal solution in just a few iterations, while previous approaches (e.g., PI-GNN) require $\sim 50$ times more iterations. Details of the computational experiment on this graph can be found in Appendix A.

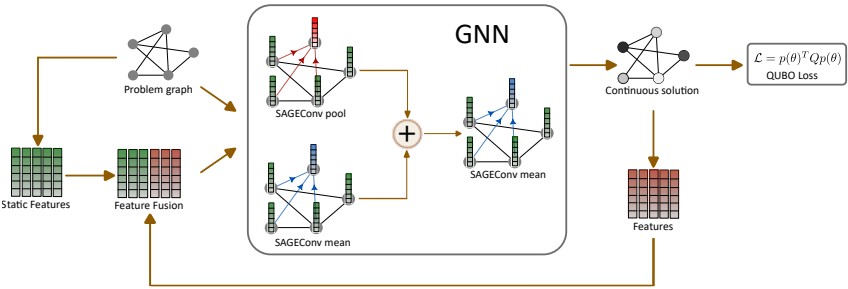

Figure 2: The QIGNN architecture. Firstly, the problem graph is associated with the initial QUBO problem, and the node static features are extracted. Then the dynamic features from the previous iteration is concatenated with the static features. Finally, these fused input vectors along with the graph data pass through the graph neural network to update probabilities $p_i$ in Eq. 3.

**Iterative design.** Let us now formalize the described intuition into an algorithm. We consider an undirected graph $G(V, E)$ with predefined static features of vertices $a_i$. At the zero step state vectors of predictions are initialized with zeros or random numbers. Together with vectors $a_i$ they constitute initial vectors $h_i^{0,0}$, where upper indices now correspond to the current optimization step and the number of a GNN layer. In the proposed approach, at each iteration $t + 1$ the output state vector $h_i^{t,N_l}$ of the node $i$ from previous iteration $t$ is used as a dynamic node feature.

Thus, the combination of static $a_i$ and dynamic features is treated as input of the graph neural network (line 5 in Algorithm 1). The resulting vectors $h_i^{t+1,0}$ for each node $i$ then engage in convolutions of GNN (line 7-12). Model weights are updated according to the loss function at each iteration $t$, so the matrix of parameters $W^{t+1,l}$ of GNN layer $l$ is dependent on the previous state $h_i^{t,N_l}$ (line 13-14). It is also possible to use the probabilities (Equation 3 and line 16 in Algorithm 1) as the dynamic part, but we chose the output state vectors $h_i^{t,N_l}$ motivated by the difference in quality of solutions in the experiments. The symbol $\sigma$ refers to the activation function used in the output layer to get probabilities. This is a sigmoid for binary prediction tasks (Max-Cut and MIS), and a softmax for multi-class prediction (Graph Coloring).

In this design, the time sequence of states is limited only by the number of iterations and convergence criteria, and optimization is performed for the current distribution of node states. The neural network predicts next states of vertices within an update, taking into account the relative distribution of states of neighbouring vertices obtained at iteration $t$ and the initial static features of nodes. The proposed method allows to change the current state of a node closer to the optimal one and to explore the state space widely due to the dynamic part of feature vectors.

## 3.2 ARCHITECTURE AND TRAINING PIPELINE

The proposed framework admits the use of different graph convolution layers. In this work, three options were explored, namely GATv2 Brody et al. (2021), GCN Kipf and Welling (2017) and SAGE Hamilton et al. (2017) convolutional layers. We have shown that results were improved in all cases, but the SAGE convolution attained better performance and was therefore chosen for the main experiments (please see the ablation study 5 for more details).

To improve the representative power of GNNs, we suggest the use of parallel layers, which represents multi-level feature extraction similar to Inception module from computer vision field by Szegedy et al. (2015). Final architecture consists of three SAGE convolutions with different types of aggregation (see Figure 2 and detailed description in Appendix A).

Mean and pooling aggregation functions were chosen for two parallel intermediate SAGE layers, and the mean aggregation function was chosen for the last SAGE layer. The pool aggregation allows to determine the occurrence

---

**Algorithm 1** Incorporation of the predicted solution into the iterative optimization process

1: **Input:** Graph $G(V, E)$, features $\{a_i, \forall i \in V\}$
2: **Output:** Class probabilities $\{p_i, \forall i \in V\}$
3: **for** $t \in \{0, \ldots, N_t - 1\}$ **do**
4:    **for** $i \in V$ **do**
5:       $h_i^{t+1,0} \leftarrow [a_i, h_i^{t,N_l}]$
6:    **end for**
7:    **for** $l \in \{0, \ldots, N_l - 1\}$ **do**
8:       **for** $i \in V$ **do**
9:          $h_{N(i)}^{t+1,l+1} \leftarrow \rho\left(\left\{h_j^{t+1,l}, \forall j \in N(i)\right\}\right)$
10:       $h_i^{t+1,l+1} \leftarrow f\left(W^{t,l}\left[h_i^{t+1,l} \; h_{N(i)}^{t+1,l+1}\right]\right)$
11:       **end for**
12:    **end for**
13:    $\mathcal{L} \leftarrow \mathcal{L}\left[\sigma\left(\left\{h_i^{t+1,N_l}, \forall i \in V\right\}\right)\right]$
14:    $W^{t+1,l} \leftarrow W^{t,l} - \gamma \nabla_{\theta^l} \mathcal{L}, \forall l$
15: **end for**
16: $\{p_i \leftarrow \sigma(h_i^{N_t, N_l}), \forall i \in V\}$

---

of certain classes among features of neighboring nodes. The mean aggregation shows the ratio of the number of different classes in the neighborhood of the considered node. This architecture configuration with a small number of successive layers allows to store information about local neighborhoods without much over-smoothing Rusch et al. (2023), and its advantages are supported by the ablation study in Appendix B. Hereafter, the method based on QUBO loss minimization by iterative solution refinement using the GNN with the described architecture will be referred to as QIGNN.

As mentioned above, there are several methods how to generate static input features $a_i$, which then go through a neural network. In this work, we create a static feature vector as a composite vector of a random part, shared vector Cui et al. (2022) and pagerank Brin and Page (1998). At the first iteration, the probability vector $h_i^{0,0}$ is initialized with zeros. Another way involves one-hot encoding for each node and then training a special embedding layer as in the PI-GNN Schuetz et al. (2022a) architecture. It allows the neural network itself to learn the most representative features. However, we do not use an embedding layer, since it requires additional computational resources and has shown no benefit over artificial features within the framework of conducted experiments (see Appendix B).

# 4 NUMERICAL EXPERIMENTS

We evaluate QIGNN on three canonical combinatorial optimization problems: Max-Cut, Maximum Independent Set (MIS), and Graph Coloring. In each section, we provide a QUBO formulation of the problem, details of the experiments, and computational results. The selected benchmarks include synthetic random graphs and real-world instances of varying structure. We compare QIGNN against state-of-the-art learning methods and classical heuristics selected per problem.

We found that, unlike PI-GNN, QIGNN does not require per-instance hyperparameter tuning to outperform prior GNNs. While tuning may yield marginal gains, we fix most parameters across experiments: we vary only the hidden layer size and iteration count, as some problems converge with fewer steps. Since QIGNN is sensitive to initialization, we run multiple seeds and report the best result. Full experimental setup details are provided in Appendix A.

## 4.1 MAXIMUM CUT

**QUBO Formulation.** The Max-Cut problem involves partitioning the vertices $V$ into two subsets such that the number of edges with endpoints in different subsets is maximized (or the total weight of such edges in the case of a weighted graph). Its QUBO objective function is as follows:

$$\mathcal{F}(x) = \sum_{i<j} A_{ij}(2x_i x_j - x_i - x_j), \quad \forall i \in V \tag{4}$$

where $A$ is an adjacency matrix of $G$. The decision variable $x_i = 1$ indicates that vertex $i$ is in one subset, while $x_i = 0$ indicates that it belongs to another one. In this paper we consider only unweighted graphs with $A_{ij} = 1, \forall (i, j) \in E$.

**Experimental Setup and Numerical Results.** We compare QIGNN against operations research (OR) methods, unsupervised learning algorithms (UL) and heuristics (H) on BA graphs with 4 attaching edges Barabási and Albert (1999), random regular graphs and the Gset benchmark Ye (2003). We evaluate the performance, the inference time and report the mean value of the number of cuts and the approximation ratio relative to the best-performing non-ML solver Gurobi, following Zhang et al. (2023) (see Table 1).

Table 1: Max-Cut results on small (|V| between 200 and 300) and large (|V| between 800 and 1200) BA graphs sets. Column "SIZE" corresponds to the average number of cuts, "DROP" shows performance drop ratio in comparison with the Gurobi method, and inference time is shown in the form of hour:minute:second or minute:second.

| METHOD | TYPE | BA-[200-300] | | | BA-[800-1200] | | |
|---|---|---|---|---|---|---|---|
| | | SIZE ↑ | DROP ↓ | TIME ↓ | SIZE ↑ | DROP ↓ | TIME ↓ |
| GUROBI | OR | **732.47** | 0.00% | 13:04 | 2915.29 | 0.00% | 1:05:29 |
| SDP | OR | 700.36 | 4.38% | 35:47 | 2786.00 | 4.43% | 10:00:00 |
| GREEDY | H | 688.31 | 6.03% | 0:13 | 2761.06 | 5.29% | 3:07 |
| MFA | H | 704.03 | 3.88% | 1:36 | 2833.86 | 2.79% | 7:16 |
| ERDOS | UL | 693.45 | 5.33% | 0:46 | 2870.34 | 1.54% | 2:49 |
| ANNEAL | UL | 696.73 | 4.88% | 0:45 | 2863.23 | 1.79% | 2:48 |
| GFLOWNET | UL | 704.30 | 3.85% | 2:57 | 2864.61 | 1.74% | 21:20 |
| QIGNN | UL | **732.2** | 0.04% | 8:12 | **2965.85** | -1.71% | 15:10 |

The results are averaged across 500 random BA graphs for each category, i.e. the small graph set with number of vertices between 200 and 300 and large graph set with number of vertices between 800 and 1200. As can be seen, on the large-graph set, QIGNN outperforms all the approaches considered while being several times faster than the gold standard, Gurobi. On the small-graph set, QIGNN is comparable in quality to Gurobi and also shows better performance than the others. The results for random regular graphs can be found in Appendix A.3, where the advantage of the proposed algorithm is also demonstrated on various sets of random graphs.

Table 2 compares QIGNN with PI-GNN, RUN-CSP, EO, and two state-of-the-art heuristics: Breakout Local Search (BLS) by Benlic and Hao (2013) and the Hybrid Evolutionary Algorithm (TSHEA) by Wu et al. (2015). We reimplemented the $\tau$-EO heuristic based on Boettcher and Percus (2001) (details in Appendix A). QIGNN, BLS, and TSHEA were each run 20 times per graph; for RUN-CSP, the best of 64 runs was selected. PI-GNN results were obtained using graph-specific hyperparameter tuning as described in Schuetz et al. (2022a). Our QIGNN outperforms both neural methods and the EO heuristic. While BLS and TSHEA perform better on smaller instances, QIGNN achieves the best cut on the largest graph (G70) in ∼1000 seconds—an order of magnitude faster than BLS (∼11,000s) and TSHEA (∼7000s) Benlic and Hao (2013); Wu et al. (2015).

## 4.2 GRAPH COLORING

**QUBO Formulation.** We consider two variants of the graph coloring problem. In the first, the graph $G = (V, E)$ must be colored using colors $k$ to minimize violations, that is, adjacent nodes that share the same color. The second formulation seeks the smallest $k$ for which a valid coloring exists. The

Table 2: Number of cut edges for benchmark instances from Gset Ye (2003) with the number of nodes $|V|$ and the number of edges $|E|$. QIGNN outperforms all other GNN-based approaches and the EO heuristic, while being comparable to SOTA heuristics.

| GRAPH | $|V|$ | $|E|$ | HEURISTICS | | | UNSUPERVISED LEARNING | | |
|---|---|---|---|---|---|---|---|---|
| | | | **BLS** | **TSHEA** | **EO** | **PI-GNN** | **RUN-CSP** | **QIGNN** |
| G14 | 800 | 4694 | **3064** | **3064** | 3058 | 3026 | 2943 | 3058 |
| G15 | 800 | 4661 | **3050** | **3050** | 3046 | 2990 | 2928 | 3049 |
| G22 | 2000 | 19990 | **13359** | **13359** | 13323 | 13181 | 13028 | 13340 |
| G49 | 3000 | 6000 | **6000** | **6000** | **6000** | 5918 | **6000** | **6000** |
| G50 | 3000 | 6000 | **5880** | **5880** | 5878 | 5820 | **5880** | **5880** |
| G55 | 5000 | 12468 | 10294 | **10299** | 10212 | 10138 | 10116 | 10282 |
| G70 | 10000 | 9999 | 9541 | 9548 | 9433 | 9421 | 9319 | **9559** |

QUBO objective in both problems is formulated as follows:

$$\mathcal{F}(x) = \sum_i \left(1 - \sum_c x_{i,c}\right)^2 + \sum_{(i,j)\in E} \sum_c x_{i,c} x_{j,c}, \quad \forall i \in V, \ \forall c \in \{1, \ldots, k\},$$

where $k$ is the number of colors the graph has to be colored. In the second variant, $k$ is not fixed and is part of the optimization. The loss function can be reduced to the second term of the objective in order to train GNN, because the softmax output enforces that each node selects a unique color, implicitly satisfying the first constraint.

**Experimental Setup and Results.** We evaluate QIGNN on synthetic graphs from the COLOR dataset Trick (2002) and three real-world citation graphs: Cora, Citeseer, and Pubmed Li et al. (2022). The synthetic graphs contain 25–561 nodes, while citation graphs scale up to 20,000. Full graph specifications are in Appendix A.4. We compare QIGNN against GNN baselines PI-GNN by Schuetz et al. (2022b), GNN-1N by Wang et al. (2023), GDN by Li et al. (2022) and RUN-CSP by Tönshoff J and M (2021), and the SOTA heuristics HybridEA by Galinier and Hao (1999). HybridEA results were obtained using the implementation [1] based on Lewis (2021). We performed up to 10 runs for select instances; most graphs required only one. Table 3 shows the best result of algorithms for coloring a graph with a chromatic number of colors. A violation is counted when adjacent nodes share the same color. QIGNN achieves the best results on all instances, outperforming SOTA HybridEA on several graphs. Table 4 shows the number of colors that the algorithm needs to color the graph without violations. As GNN-1N and GDN do not report results for this formulation, we omit them

Table 3: The number of violations when coloring the graph with chromatic number of colors by HybridEA (HEA) heuristics and GNN-based methods GNN-1N, PI-GNN, GDN and QIGNN for citation graphs and graphs from the COLOR dataset.

| GRAPH | HEUR | UNSUPERVISED LEARNING | | | |
|---|---|---|---|---|---|
| | **HEA** | **GNN-1N** | **PI-GNN** | **GDN** | **QIGNN** |
| HOMER | 0 | 0 | 0 | 0 | 0 |
| MYCIEL6 | 0 | 0 | 0 | 0 | 0 |
| QUEEN5-5 | 0 | 0 | 0 | 0 | 0 |
| QUEEN6-6 | 0 | 0 | 0 | 0 | 0 |
| QUEEN7-7 | 0 | 0 | 0 | 9 | 0 |
| QUEEN8-8 | 0 | 1 | 1 | - | 0 |
| QUEEN9-9 | 0 | 1 | 1 | - | 0 |
| QUEEN8-12 | 0 | 0 | 0 | 0 | 0 |
| QUEEN11-11 | 14 | 13 | 17 | 21 | **7** |
| QUEEN13-13 | 18 | 15 | 26 | 33 | **15** |
| CORA | 0 | 1 | 0 | 0 | 0 |
| CITESEER | 0 | 0 | 0 | 0 | 0 |
| PUBMED | 0 | - | 17 | 21 | **0** |

Table 4: The number of color needed for coloring without violations by HybridEA (HEA) heuristics and GNN-based methods PI-GNN, RUN-CSP and QIGNN on citation graphs and graphs from the COLOR dataset. Here $\chi$ is a known chromatic number.

| GRAPH | $\chi$ | HEUR | UNSUPERVISED LEARNING | | |
|---|---|---|---|---|---|
| | | **HEA** | **PI-GNN** | **RUN-CSP** | **QIGNN** |
| HOMER | 13 | **13** | **13** | 17 | **13** |
| MYCIEL6 | 7 | **7** | **7** | 8 | **7** |
| QUEEN5-5 | 5 | **5** | **5** | 5 | **5** |
| QUEEN6-6 | 7 | **7** | **7** | 8 | **7** |
| QUEEN7-7 | 7 | **7** | **7** | 10 | **7** |
| QUEEN8-8 | 9 | **9** | 10 | 11 | **9** |
| QUEEN9-9 | 10 | **10** | 11 | 17 | **10** |
| QUEEN8-12 | 12 | **12** | **12** | 17 | **12** |
| QUEEN11-11 | 11 | **12** | 14 | >17 | **12** |
| QUEEN13-13 | 13 | **14** | 17 | >17 | 15 |
| CORA | 5 | **5** | **5** | - | **5** |
| CITESEER | 6 | **6** | **6** | - | **6** |
| PUMBED | 6 | **8** | 9 | - | **8** |

[1] http://rhydlewis.eu/gcol/

and include RUN-CSP for comparison. For QIGNN, we successively increased the number of colors to find the optimal one. In this setting, QIGNN outperforms all GNN baselines and is outscored by HybridEA on only one instance.

### 4.3 Maximum Independent Set

**QUBO Formulation.** For a given graph $G = (V, E)$ the Maximum Independent Set (MIS) problem is to find a subset $S \subset V$ of pair-wise nonadjacent nodes of the maximum size $|S|$. The QUBO cost function of MIS is:

$$\mathcal{F}(x) = -\sum_{i \in V} x_i + P \sum_{(i,j) \in E} x_i x_j \quad \forall i \in V. \tag{5}$$

where $x_i = 1$ if node $i \in S$ and $P$ is a penalty coefficient for violating the independence condition, which ensures that no two adjacent nodes are both included in the independent set. Unlike the previously considered problems, this formulation contains a penalty term, raising the question of which value of $P$ should be chosen. In our algorithm an adaptive coefficient was applied. For more details, please see Appendix A.

**Experimental Setup and Numerical Results.** For MIS, we conduct experiments on randomly generated graphs of different structures. Table 5 compares QIGNN with state-of-the-art learning-based methods on two Erdos–Rényi (ER) graph sets: 500 instances each with 700–800 and 9000–11,000 nodes Erdos and Rényi (1984). We compare QIGNN to state-of-the-art supervised methods: Intel by Li et al. (2018), Difusco by Sun and Yang (2023) and T2TCO by Li et al. (2023); LwD by Ahn et al. (2020) and DIMES by Qiu et al. (2022) are recent Reinforcement Learning (RL) methods, GflowNets by Zhang et al. (2023) We also include the classical MIS solver KaMIS (ReduMIS) Lamm et al. (2017) as the strongest non-learning baseline. When multiple decoding strategies (e.g., greedy or sampling) are reported in sources, we select the variant with the best average MIS size. Difusco and T2TCO could not scale to the ER-9000–11000 dataset Zhang et al. (2023), and are therefore omitted. As shown in Table 5, QIGNN outperforms all learning-based baselines. The gap is most pronounced on large graphs, where QIGNN even surpasses KaMIS while running significantly faster.

Table 5: Comparison of average found MIS sizes and runtime for QIGNN, learning-based methods and the SOTA heuristics KaMIS on sets of 500 Erdos-Renyi random graphs with a different number of nodes.

| Method | Type | ER-[700-800] | | ER-[9000-11000] | |
|---|---|---|---|---|---|
| | | Size | Time | Size | Time |
| KaMIS | Heur | 44.87 | 52:13 | 374.57 | 7:37:21 |
| Intel | SL | 34.86 | 6:04 | 284.63 | 5:02 |
| Difusco | SL | 40.35 | 32:98 | - | - |
| T2TCO | SL | 41.37 | 29:44 | - | - |
| LwD | RL | 41.17 | 6:33 | 345.88 | 1:02:29 |
| DIMES | RL | 42.06 | 12:01 | 332.8 | 12:31 |
| GflowNets | UL | 41.14 | 2:55 | 349.42 | 1:49:43 |
| CRA | UL | 41.64 | 47.30 | 360.71 | 1:03:00 |
| QIGNN | UL | **42.45** | 3:46 | **375.44** | 10:32 |

Following Tönshoff J and M (2021), we also evaluate on RB-model graphs Xu and Li (2006), which are hard instances with hidden optima ( Table 6). On this benchmark, QIGNN outperforms RUN-CSP and greedy baselines, and comes close to KaMIS.

Table 6: MIS results of QIGNN, GNN-based method RUN-CSP, greedy and SOTA KaMIS heuristics on RB Model graphs. We report average MIS sizes over 5 runs with standard deviations.

| Graph | $|V|$ | $|E|$ | Heuristics | | Unsupervised Learning | |
|---|---|---|---|---|---|---|
| | | | KaMIS | Greedy | RUN-CSP | QIGNN |
| FRB30-15 | 450 | 18k | $30 \pm 0.0$ | $24.6 \pm 0.5$ | $25.8 \pm 0.8$ | $28.4 \pm 0.4$ |
| FRB40-19 | 790 | 41k | $39.4 \pm 0.5$ | $33.0 \pm 1.2$ | $33.6 \pm 0.5$ | $36.8 \pm 0.7$ |
| FRB50-23 | 1150 | 80k | $48.8 \pm 0.4$ | $42.2 \pm 0.8$ | $42.2 \pm 0.4$ | $45 \pm 0.6$ |
| FRB59-26 | 1478 | 126k | $57.4 \pm 0.9$ | $48.0 \pm 0.7$ | $49.4 \pm 0.5$ | $54.6 \pm 1$ |

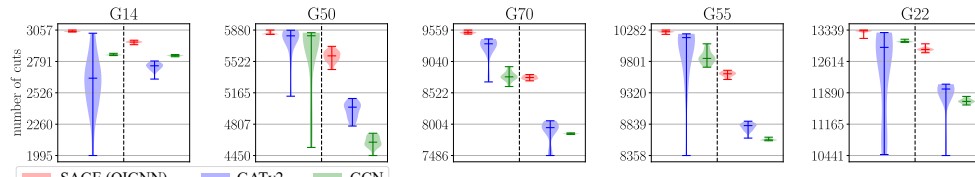

Figure 3: Distribution of the results over 20 runs of GNNs with different convolutional layers with (left) and without (right) the iterative approach on several instances from Gset. Horizontal lines mean maximum, median and minimum values.

## 5 IMPACT OF THE ITERATIVE REFINEMENT ON GNNs PERFORMANCE.

We have investigated how the iterative refinement of distribution of vertex class membership affects performance of GNN architectures with different types of convolutions, namely GCN Kipf and Welling (2017) and GATv2 Brody et al. (2021) in addition to the default architecture of QIGNN with SAGE convolutions. For this purpose, we constructed GNNs from two consecutive convolutions of a given type. The sizes of hidden states were the same for all networks and coincided with the sizes of the described default QIGNN architecture. We performed 20 calculations with different seeds for graphs from Gset to solve the Max-Cut problem. Figure 3 shows that the iterative refinement proposed in this paper greatly improves the maximum cut found for all types of convolutions. At the same time, the default QIGNN shows better results compared to other architectures.

## 6 RELATED WORK

Graph neural networks are rapidly gaining popularity as a powerful tool for solving CO problems Cappart et al. (2023). Supervised learning based approaches are commonly applied Prates et al. (2019); Gasse et al. (2019); Li et al. (2018); Sun and Yang (2023); Li et al. (2023). However, the need to collect labeled training instances into representative and unbiased dataset is a limitation of supervised algorithms, and often face challenges with generalization to new and unseen problem instances. Reinforcement learning presents an alternative by generating iterative solutions Khalil et al. (2017); Kool et al. (2019); Qiu et al. (2022); Darvariu et al. (2024). RL methods may experience difficulties when facing large scale problems due to the vastness of the state space, and the need of a large number of samplings. The unsupervised learning paradigm, where solvers do not require a training set of pre-solved problems, has the potential to overcome these limitations. Tönshoff J and M (2021) proposed RUN-CSP as a recurrent GNN to solve maximum constraint satisfaction. Amizadeh et al. (2018) developed GNN to solve SAT and CircuitSAT. Karalias and Loukas (2020) train GNN to obtain a distribution of nodes corresponding to the candidate solution and Sun et al. (2023) provided an annealed version of it. Wang et al. (2022) study entry-wise concave relaxations of CO objectives. Schuetz et al. (2022a); Ichikawa (2024) apply relaxed QUBO as instance specific GNN loss, Schuetz et al. (2022b); Wang et al. (2023) extend it for solving graph coloring problem.

## 7 CONCLUSION

In this work, we propose the novel iterative approach for solving combinatorial optimization problems using graph neural networks in unsupervised mode. We show that this method significantly enhances the performance of all types of GNN convolutions considered. We also suggest the design of the GNN architecture, a set of necessary features of the graph vertices, and reveal the quality in experiments on the well-known maximum cut, graph coloring and maximum independent set problems. The results of our comparative analysis demonstrate that the proposed algorithm drastically outperforms all learning-based baselines, including SOTA supervised, unsupervised, and reinforcement learning methods. Moreover, we show that it competes with the best classical heuristics for the problems addressed while showing a distinct advantage in computational time on large graphs. For the future work, we consider the algorithm superior performance and scalability promising to be extended to other CO problem formulated as QUBO, thus highlighting its potential in the field of combinatorial optimization.

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

# A  TECHNICAL DETAILS AND CONVERGENCE

## A.1  GENERAL EXPERIMENTAL SETUP

All random graphs in this work were generated by the NetworkX[2] package. To implement the QIGNN architecture the DGL library[3] was used. The pseudocode of one iteration with the QIGNN architecture is presented in the Algorithm 2. We used the Adam optimizer without a learning rate schedule. The learning rate was set empirically to 0.014, the rest of the optimizer parameters remained set by default. Gradients were clipped at values 2 of the Euclidean norm.

We limited the number of iterations to $5 \times 10^4$ for random regular graphs and $10^5$ for all the other graphs, but in some cases convergence was reached much earlier. If the value of the loss function at the last 500 iterations had differed by less than $10^{-5}$ it was decided that the convergence was achieved and the training was stopped. In the case of the graph coloring problem, an additional stopping criterion was used and the solution was considered to be found when the absolute value of the loss function becomes less than $10^{-3}$.

The dropout was set to 0.5. The dimension of the random part of input vectors was equal to 10, the size of hidden layers was fixed at 50 for Max-Cut and at 140 for graph coloring.

Due to the stochasticity of the algorithm, it is preferable to do multiple runs with different seeds to find the best result. One can do separate runs in parallel possibly utilizing several GPUs. If the device has enough memory, the RUN-CSP scheme by Tönshoff J and M (2021) can be used. In this case, one composite graph with duplicates of the original one is created for the input. We trained the model in parallel on the NVIDIA Tesla V100 GPU. Conventional heuristics were launched on the machine with two Intel Xeon E5-2670 v3 @ 2.30GHz.

We tested three ways to recursively utilize the probability data. Specifically, we passed raw probability data taken before the sigmoid layer, data after the sigmoid layer or concatenated both of these options. Different iterative dynamic features led to a minor improvement on some graphs, while at the same time slightly worsening the results on other graphs. In this work we presented results for the concatenated data.

---

**Algorithm 2** Forward propagation of the QIGNN algorithm at iteration $t$

---

1: **Input:** Graph $G(V, E)$, static nodes features $\{a_i, \forall i \in V\}$
2: **Output:** Probability $p_i$, hidden state $h_i^t, \forall i \in V$
3: $h_i^{t,0} \leftarrow \begin{bmatrix} a_i & h_i^{t-1} \end{bmatrix}, \quad \forall i \in V$
4: **for** $i \in V$ **do**
5: $\quad h_{N(i)}^{t,1} \leftarrow \rho_{\text{mean}}\left(\left\{h_j^{t,0}, \forall j \in N(i)\right\}\right)$
6: $\quad h_i^{t,1} \leftarrow f\left(W^1\left[h_i^{t,0} \; h_{N(i)}^{t,1}\right]\right)$
7: $\quad h_{N(i)}^{t,2} \leftarrow \rho_{\text{pool}}\left(\left\{h_j^{t,0}, \forall j \in N(i)\right\}\right)$
8: $\quad h_i^{t,2} \leftarrow f\left(W^2\left[h_i^{t,0} \; h_{N(i)}^{t,2}\right]\right)$
9: **end for**
10: $\{h_i^{t,1}\} \leftarrow \text{BN}_{\gamma 1, \beta 1}(\{h_i^{t,1}, \forall i \in V\})$
11: $\{h_i^{t,2}\} \leftarrow \text{BN}_{\gamma 2, \beta 2}(\{h_i^{t,2}, \forall i \in V\})$
12: **for** $i \in V$ **do**
13: $\quad h_i^{t,12} \leftarrow f(h_i^{t,1} + h_i^{t,2})$
14: $\quad h_i^{t,12} \leftarrow \text{Dropout}(h_i^{t,12})$
15: $\quad h_{N(i)}^{t,\text{out}} \leftarrow \rho_{\text{mean}}\left(\left\{h_j^{t,12}, \forall j \in N(i)\right\}\right)$
16: $\quad h_i^{t,\text{out}} \leftarrow f\left(W^{\text{out}}\left[h_i^{t,12} \; h_{N(i)}^{t,\text{out}}\right]\right)$
17: **end for**
18: $p_i, h_i^t \leftarrow \sigma(h_i^{t,\text{out}}), h_i^{t,\text{out}}, \quad \forall i \in V$

---

[2]https://networkx.org/
[3]https://www.dgl.ai/

## A.2 Convergence

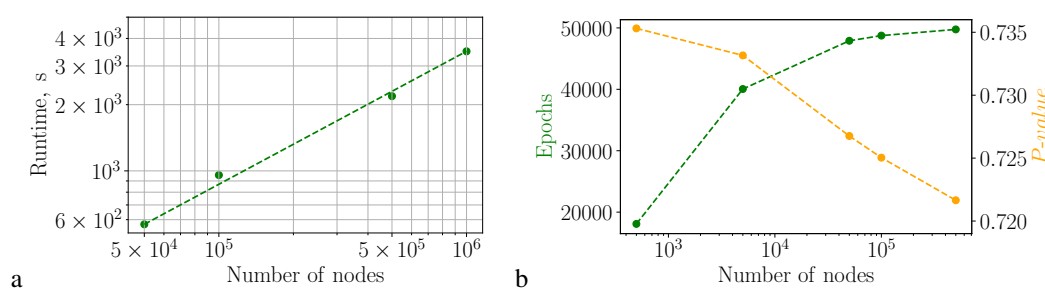

Figure 4: a) The computation time of $5 \times 10^4$ iterations of QIGNN on random regular graphs with $d = 5$ in the sparse format depending on the number of nodes. b) The iteration number averaged over 20 graphs at which the algorithm found the best solution during the training process (green) and the mean *P-value* for 1 run (orange).

The number of runs and iterations in experiments were not optimal and were chosen for a more fair comparison with other algorithms. More runs and iterations can lead to better results. We conducted additional experiments on 20 random regular graphs with $d = 5$ and up to one million nodes. One run was made for each graph and the number of iterations was limited to $5 \times 10^4$. The training time for large graphs in sparse format on single GPU is shown in Figure 4a. We also analyzed how the number of iterations can affect the quality of the solution. As the number of vertices increases, the iteration at which the last found best solution was saved moves closer to the specified boundary (see Figure 4b). Meanwhile, the average *P-value* of one run drops from $0.735$ to $0.722$ and one of the reasons for this may include the limited duration of training. If, for example, we train QIGNN for $10^5$ iterations on graphs with $n = 5 \times 10^4$ nodes, the average *P-value* will increase from $0.726$ to $0.728$, while on small graphs with $n = 500$ we do not observe such an effect. Thus, it is difficult to talk about the convergence of the algorithm on large instances under the given constraint. The recommendation is to follow the latest best solution updates and terminate the algorithm if it does not change for a sufficiently large ($> 10^4$) number of iterations.

In order to study the robustness of the algorithm with respect to changes in hyperparameters, we run the default QIGNN architecture with two parallel layers on graphs from the Gset dataset for the Max-Cut problem. All hyperparameters except the learning rate were chosen as described in Section 4. As can be seen from Figure 5, small values of the learning rate do not allow to achieve convergence in $10^5$ iterations. For the learning rate greater than $0.01$ the results become relatively stable and the best number of cuts is achieved for values from $0.01$ to $0.02$.

## A.3 Max-Cut

For synthetic instances, the number of cut edges serves as the evaluation metric. For random regular graphs with the number of vertices $n \to \infty$, there exists a theoretical estimate of the maximum cut size that depends on $n$, vertex degree $d$ and a universal constant $P_* \simeq 0.7632$. Such asymptotics motivated the metric *P-value* $= \sqrt{\frac{4}{d}} \left( \frac{z}{n} - \frac{d}{4} \right)$, where $z$ corresponds to the obtained cut size (see Yao et al. (2019)). Higher P-values indicate better cuts, with $P_*$ serving as an upper bound on optimality.

Table 7 reports average P-values over 200 random $d$-regular graphs with $n = 500$ for QIGNN, RUN-CSP, and extremal optimization (EO) Boettcher (2003) baselines. For QIGNN and PI-GNN, we report the best result out of 5 runs. PI-GNN uses the GCN-based setup from Schuetz et al. (2022a) for $d > 5$; other values are from the original source. EO and RUN-CSP results are taken from Tönshoff J and M (2021), where RUN-CSP was evaluated using 64 runs per instance. QIGNN consistently outperforms both RUN-CSP and PI-GNN, and achieves the best performance overall for $d \geq 5$. With 15 runs, QIGNN also surpasses all baselines on $d = 3$ graphs, reaching a P-value of $0.727$.

In Table 7 results of EO and RUN-CSP were taken from Tönshoff J and M (2021), where *P-value* was averaged over 1000 graphs. RUN-CSP was allowed to make 64 runs for each graph and in the case of EO the best of two runs was chosen Yao et al. (2019). *P-values* of PI-GNN depend on the particular

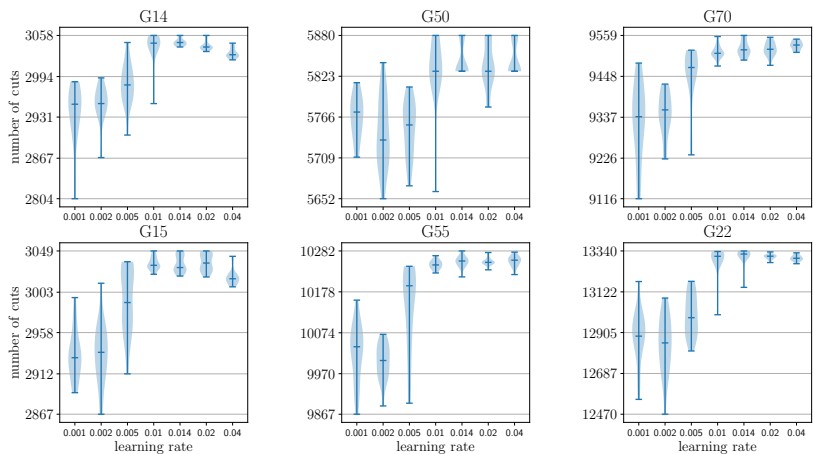

Figure 5: Results distribution for 20 runs of the default QIGNN architecture depending on the learning rate for the Max-Cut problem on several instances from Gset. The number of iterations in all cases was fixed at $10^5$. Horizontal lines mean maximum, median and minimum values.

Table 7: *P-value* of EO, PI-GNN, RUN-CSP and QIGNN for d-regular graphs with 500 nodes and different degree $d$ averaged over 200 randomly generated instances.

| **d** | HEUR | UNSUPERVISED LEARNING | | |
|---|---|---|---|---|
| | **EO** | **PI-GNN** | **RUN-CSP** | **QIGNN** |
| 3 | **0.727** | 0.612 | 0.714 | 0.725 |
| 5 | 0.737 | 0.608 | 0.726 | **0.738** |
| 10 | 0.735 | 0.658 | 0.710 | **0.737** |
| 15 | 0.736 | 0.644 | 0.697 | **0.739** |
| 20 | 0.732 | 0.640 | 0.685 | **0.735** |

architecture. Results for graphs with a degree 3 and 5 were published in Schuetz et al. (2022a) for the architecture with GCN layer, and it corresponds to the value in the column for PI-GNN. The cut size was bootstrap-averaged over 20 random graph instances and PI-GNN took up to 5 shots. In the paper by Schuetz et al. (2023) the authors considered another option with the SAGE layer and showed that in this case the results for graphs with a degree 3 can be improved by 10.78%. However, we did not notice an improvement over the GCN architecture on graphs with a higher degree.

To make the evaluation more informative, we implemented $\tau$-EO heuristic from Boettcher and Percus (2001). As suggested by authors, we set $\tau = 1.3$ and the number of single spin updates was limited by $10^7$. For small graphs $\tau$-EO can find a high-quality solution, but with increase of the graph size the accuracy of the algorithm degrades due to the limited number of updates. This behavior is expected by the authors, who suggested optimal scaling for number of updates as $\sim O(|V|^3)$. However, it is computationally expensive to carry out the required number of iterations. We performed 20 runs of EO with different initializations to partially compensate for this. Within the given limit, the EO algorithm took $\sim$6800 seconds per run to obtain a solution for the relatively large graph G70.

QIGNN as well as BLS and TSHEA was run 20 times on each graph. The best attempt out of 64 was chosen for RUN-CSP in original papers. In order to obtain the results of PI-GNN, the authors applied hyperparameter optimization for each graph. The results of RUN-CSP for the G70 graph was obtained by running the code[4] with parameters reported in Tönshoff J and M (2021).

---

[4] https://github.com/toenshoff/RUN-CSP

### A.4 COLORING

To find the number of colors required to color the graph without violations, we successively increased the number of colors in each new run until the correct coloring was found among 10 seeds. The number of nodes and edges of the investigated graphs is presented in Table 9. To evaluate the results of QIGNN, we did up to 10 runs for some graphs, although most of them required only one run. The convergence time for PI-GNN and QIGNN is shown in Table 8.

Results for graphs anna, david, games120, muciel5, huck and jean were omitted in the main tables because all algorithms find optimal solutions without violations. Results of GNN-1N for citation graphs were obtained by implementing the algorithm from the original paper. Since there was no instruction how to optimize hyperparameters, we took them close to PI-GNN and chose the best among 10 runs.

The number of iterations for convergence of QIGNN on citation graphs was no more than 6000 and varied for the COLOR dataset from $\sim200$ to $9 \times 10^4$. The estimated runtime for QIGNN turned out to be significantly less than for PI-GNN, and in some cases the difference reaches more than three orders of magnitude. This is due to the fact that QIGNN does not require exhaustive tuning of hyperparameters for each instance in contrast to PI-GNN Schuetz et al. (2022b).

### A.5 MIS

In our experiments, we found that setting a small $P$ leads to the fact that the solutions found by the algorithm for a given number of iterations contain too many violations. A large $P$ value can cause the algorithm to quickly converge to a trivial solution with zero set size. To circumvent the problem of adjusting $P$ for different types of graphs, we propose in this paper to linearly increase the penalty value from 0.01 to 2 throughout all the iterations. This allows the algorithm to start the search in the space of large sets with violations while gradually narrowing the search space towards sets without violations.

### A.6 TOY EXAMPLE OF A MAX-CUT PROBLEM

We provide a toy example of a Max-Cut problem to illustrate the performance of our proposed QIGNN method compared to the PI-GNN approach. The problem instance consists of a graph with 12 edges and 10 nodes, as shown in Figure 1.

An experimental setup for the QIGNN architecture is set by default similar to the description in section 4. The PI-GNN architecture consists of a trainable embedding layer and one hidden layer, the sizes of which are set similarly to QIGNN and equal to 50.

For the considered problem, QIGNN finds the optimal solution (cut = 12) in an average of 10.96 iterations, while PI-GNN requires 532.1 iterations on average. These results are based on 100 runs with different random seeds.

Table 8: Approximate runtime in seconds for PI-GNN and QIGNN training on a single GPU on instances from the COLOR dataset and citation graphs.

| Graph | $|V|$ | $|E|$ | $PI - GNN, \times 10^3$s | QIGNN, $\times 10^3$s |
|---|---|---|---|---|
| COLOR | 25-561 | 160-3328 | $3.6 \div 28.8$ | $0.002 \div 1$ |
| CORA | 2708 | 5429 | 0.3 | 0.06 |
| CITESEER | 3327 | 4732 | 2.4 | 0.018 |
| PUBMED | 19717 | 44338 | 24 | 0.156 |

Table 9: Number of Vertices and Edges in coloring graphs.

| Graph | $|V|$ | $|E|$ |
|---|---|---|
| ANNA | 138 | 493 |
| DAVID | 87 | 406 |
| GAMES120 | 120 | 638 |
| HOMER | 561 | 1629 |
| HUCK | 74 | 301 |
| JEAN | 80 | 254 |
| MYCIEL5 | 47 | 236 |
| MYCIEL6 | 95 | 755 |
| QUEEN5-5 | 25 | 160 |
| QUEEN6-6 | 36 | 290 |
| QUEEN7-7 | 49 | 476 |
| QUEEN8-8 | 64 | 728 |
| QUEEN9-9 | 81 | 1056 |
| QUEEN8-12 | 96 | 1368 |
| QUEEN11-11 | 121 | 1980 |
| QUEEN13-13 | 169 | 3328 |
| CORA | 2708 | 5429 |
| CITESEER | 3327 | 4732 |
| PUBMED | 19717 | 44338 |

## B  ABLATION FOR QIGNN COMPONENTS

We analyzed which components of QIGNN make the greatest contribution to its performance on the example of Max-Cut problem-solving. The default architecture includes two intermediate SAGEConv layers and the iterative dynamic feature. The most dramatic drop in quality occurs if the iterative part is excluded (see Figure 7).

Throwing out one of the intermediate convolutional layer does not result in such a strong downgrade (see Figure 8). However, the absence of the convolutional layer with a pool aggregation function leads to a decrease in the median result, upper bound, and an increase in the results dispersion for almost all graphs. Discarding the layer with a mean aggregation function can increase the median cut and even decrease the variance in some cases, but upper bounds either stay the same or decrease, even if we double the number of parameters in the remaining hidden layer (see Figure 8). Further ablation on random regular graphs shows that the absence of any convolutional layer leads to a worse result(see Figure 9). Table 10 with average results for one seed and the best of the five seeds also confirms the advantage of using a combination of two layers.

In this work, we settled on combining a random vector, shared vector Cui et al. (2022) and pagerank Brin and Page (1998) for the input feature vector by default. Fig. 6 shows the results when one of the parts (random vectors or the pagerank of nodes) was removed. In some cases, the median improves after dropping features, but the upper bound tends to only go down as the number of input features decreases. Using an embedding layer does not show any benefit over the default version.

Table 10: The first row contains the average *P-value* over 200 random regular graphs with $d = 5$ for 1 run of QIGNN with same configurations as in Figures 8 and 9. The second row shows the average *P-value* over 200 graphs when the best cut out of 5 runs is taken.

| Runs | Default | no SAGEConv (mean) | no SAGEConv (pool) |
|---|---|---|---|
| 1 | 0.734 | 0.732 | 0.725 |
| 5 | 0.738 | 0.737 | 0.734 |

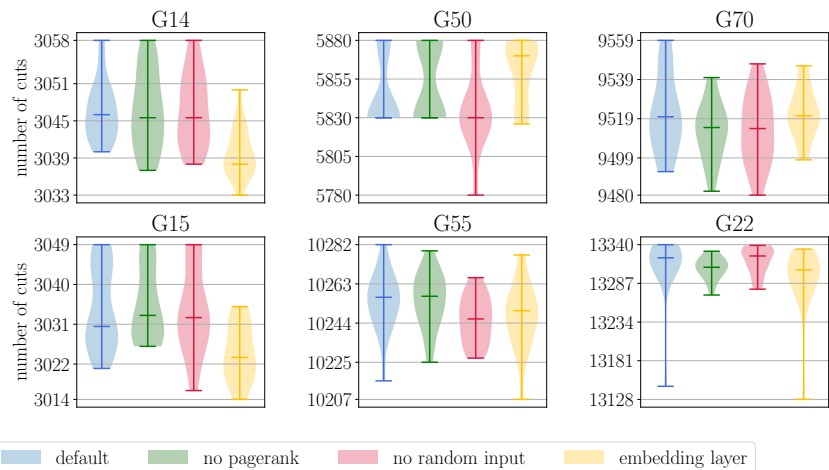

Figure 6: Results distribution for 20 runs of QIGNN with the default architecture on several instances from Gset. Input feature vectors varied as follows: the default choice corresponds to the blue color; the exclusion of the pagerank component corresponds to the green color; the exclusion of the random part corresponds to the red color. The use of a trainable embedding layer instead of artificial features is indicated in yellow.

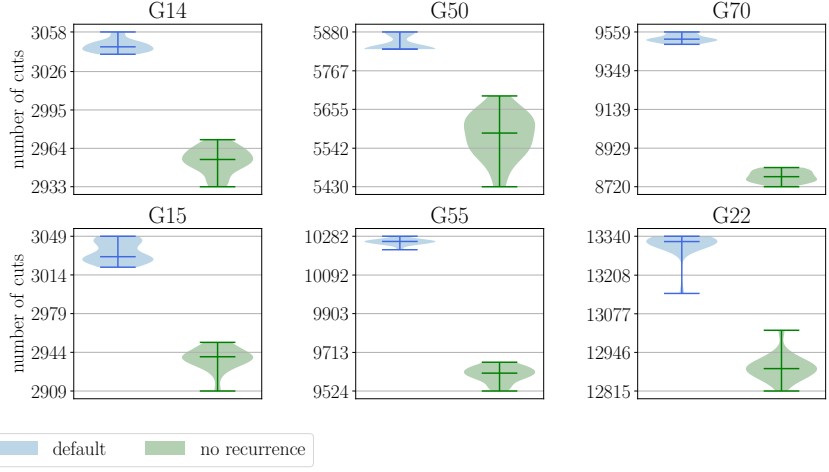

Figure 7: Results distribution for 20 runs of QIGNN with (blue) and without (green) the iterative connection on several instances from Gset. Horizontal lines mean maximum, median and minimum values.

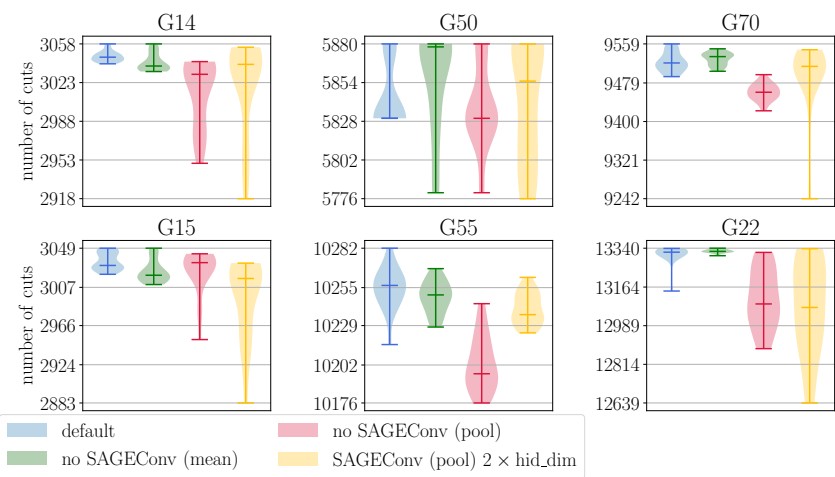

Figure 8: Results distribution for 20 runs of QIGNN of the default architecture (blue), with the absence of one SAGE layer with a mean aggregation function (green) or the SAGE layer with a pool aggregation function (red). Horizontal lines mean maximum, median and minimum values.

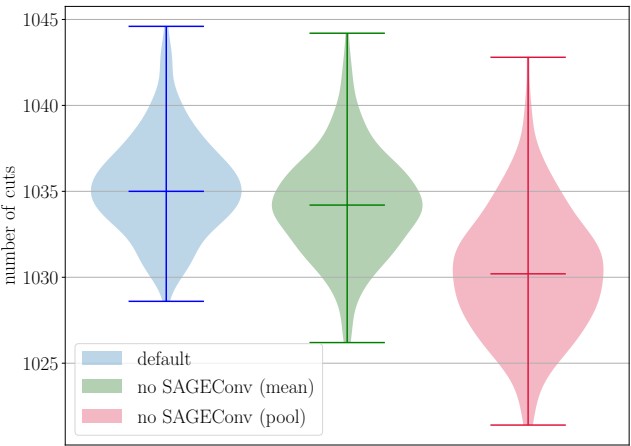

Figure 9: Results distribution for the mean of 5 runs on 200 random regular graphs with $d = 5$ and 500 nodes. QIGNN had the same configurations as in the Fig. 8.

