# OpenReview forum: "Unsupervised Graph Neural Networks for Solving Combinatorial Optimization Problems by Iterative Solution Refinement"
_ICLR.cc/2026/Conference — Submitted to ICLR 2026_

### Official Review · Reviewer_75ba · 2025-10-17

**Soundness:** 3
**Presentation:** 2
**Contribution:** 2
**Rating:** 4
**Confidence:** 3

**Summary:**

This paper introduces a novel GNN for solving several graph CO problems, max-cut, coloring, and max-independent set, reformulated as QUBO optimization problems. The key algorithmic novelty is the use of the iterative framework in the message passing layers. Results show a moderate improvement relative to baselines.

[Would have selected 5 if available. I am persuadable if you address my two concerns.]

**Strengths:**

Inclusion of dynamic features is novel

QUBO and the CO problems are well explained, and well motivated

Improvements over a fairly good set of baselines

Good ablation study

**Weaknesses:**

The iterative refinement step should be explained more clearly since this is the key algorithmic development of the paper

There is no discussion of computational cost. If the iterative method significantly increases the cost of the method, this may undercut the entire story. After all, slow exact solver exist. The name of the game is to get fast, pretty-good solvers.

Minor:

In Section 2.1, there appear to be some citep vs citet issues. (Check throughout.)
For undirected graphs it is better to write \{i,j\} rather than (i,j). (The latter is for digraphs.)
Extra space in |V | in Table 1 (twice).
Some P's and d's are incorrectly not in "math mode" in Appendix A.3

**Questions:**

Is there a good way to systematically reframe CO problems as QUBO problems (or to tell if this is possible for a given CO problem)? Or must this require "clever case-by-case thinking?"

---

> ### Author Response · Authors · 2025-11-24
> **Rebuttal by Authors**
>
> We thank the reviewer for the comments.
>
>
> **W1. The iterative refinement step explanation.**
>
> We agree that the iterative refinement step is the central contribution and should be as clear as possible. Our intent was to explain it in detail:
>
> - Lines 161–168 explain motivation of **iterative refinement**, and lines 169–180 introduce the core idea of iteratively refining a candidate solution using a **dynamic feature - a node state state from the previous epoch (iterarion)**.
> - **Figure 1** illustrates iterative refinement on a small toy example.
> - **Figure 2** shows the full architecture, highlighting how the **dynamic features** (the previous-state logits) are fed back and fused with **static features** at each iteration. Lines 196–210 provide the corresponding formal description.
> - **Algorithm 1** (lines 221–239) specifies the iterative refinement scheme step by step, including the feature update (lines 5,9,10 of the algorithm 1).
> - **Algorithm 2** in the appendix (lines 730–752) gives the full forward pass that uses the iterative design in our implementation.
>
> We would be grateful if the reviewer could indicate which specific aspects of the mechanism remained unclear, so that we can address those points directly and explain it better in the revised version of the paper.
>
> **W2. Computational cost.**
>
> We fully agree that “fast, pretty-good solvers” are the main goal. In our case, the iterative refinement step **does not  fundamentally change** the computational profile of the model: each iteration has the same complexity as a standard GNN forward pass (\(O(|E|d)\)), and the only overhead is that we process a slightly larger feature vector (we add **one dynamic feature** channel containing the previous state on top of the static features). In other words, we go from processing **\(x\)** features to **\(x+1\)**; the number of message-passing layers and the overall asymptotic cost remain the same. **Appendix Fig. 4** shows that in practice our runtime scales essentially linearly with the number of nodes.
>
> In terms of wall-clock time, the iterative refinement does **not** turn QIGNN into a “slow exact solver”. On the contrary class, for example even compared to PI-GNN [1] we are much faster because PI-GNN requires expensive hyperparameter optimization per task, whereas QIGNN uses a single configuration across benchmarks (see Fig. 8). In Tables 1 and 5 we also report runtimes: for example, on very large MIS instances with 9,000–11,000 nodes, QIGNN reaches better solutions in about 10 minutes, while KaMIS (one of the strongest classical solvers) takes on the order of 7 hours on average. Thus, QIGNN achieves high-quality solutions on large graphs with runtimes that are far below exact solvers being competitive with (or better than) state-of-the-art heuristics.
>
> We thank the reviewer for these formatting and notation suggestions and will correct them in the revised version.
>
> **Q1.** In theory, there *is* a fairly systematic route: most combinatorial optimization problems can be written as 0–1 integer (or pseudo-boolean) programs, and any such formulation can be converted to a QUBO by introducing binary variables and quadratic penalty terms for the constraints. In that sense, the question is usually answered at the level of the ILP model: if you have a 0–1 ILP, there is a standard mechanical way to obtain an equivalent QUBO.
>
> In practice, however, **good** QUBO formulations (compact, numerically well-behaved, with reasonable penalty scales) still require some problem-specific modeling choices. That is why in our paper we rely on standard, well-known QUBO encodings for Max-Cut, MIS, and Coloring rather than claiming an automatic QUBO generator.  For more details on QUBO modeling please refer to [2,3]
>
>
>
> [1] Combinatorial optimization with physics-inspired graph neural networks, 2022
>
> [2] Quantum Bridge Analytics I: A Tutorial on Formulating and Using QUBO Models, 2018
>
> [3] Ising formulations of many NP problems, 2014

---

### Official Review · Reviewer_BTQ4 · 2025-10-30

**Soundness:** 2
**Presentation:** 3
**Contribution:** 2
**Rating:** 6
**Confidence:** 3

**Summary:**

The paper proposes an instance-wise, unsupervised GNN framework for QUBO-formulated combinatorial optimization. The main novelty is iterative refinement: the previous iteration’s final hidden state is fed to the first layer at the next iteration. Benchmarks are reported on Max-Cut, Graph Coloring, and MIS. The paper claims better performance than other learning-based methods and than SOTA heuristics on larger instances.

**Strengths:**

The manuscript appears to have undergone substantial editing phases.
The introduction of the method is clear. Benchmarks are broad across three CO tasks with comparisons against numerous other approaches. A simple modification to a conventional GNN yields a significant effect, which is appealing.
The claim that it works better on larger instances is interesting if substantiated.

**Weaknesses:**

There is an accumulation of benchmark results with different solver sets and unclear reasoning for the choice of baselines.
This makes it hard to understand the true potential of the method (see questions for details).

**Questions:**

1) In principle, iterating may require backpropagation through time when differentiating the loss. Could the author clarify: is the previous state explicitly detached and is no BPTT used? If so, it is important to state this clearly or justify why there is no need to consider BPTT.

2) Why does Table 1 not show results for stronger heuristics such as EO (if EO is already re-implemented)? The baseline heuristics in this table are quite weak (MFA and Greedy).

3) For Table 2: the apparent win is driven by a single instance (G70). On G14, G15, G22, G49, G50, and G55, the proposed method is at best comparable and often below BLS/TSHEA. Please explain why G70 improves while G55 does not. Is this due to graph-type differences (e.g., degree or density) or just size? It would help to report per-instance runtime for all methods on the same hardware.

4) The same comments apply to Tables 3 and 4.

5) The paper hints that the advantage occurs at large problem size. It would be clearer to show scaling versus problem size on an artificial ensemble of instances using a time-to-target metric, comparing to SOTA or strong heuristics. This would make a much stronger case that the proposed method works better on larger graphs.

6) It is stated that “Difusco and T2TCO could not scale to the ER-9000–11000 dataset,” but the proposed method could. Is this due to memory cost? If so, it would be interesting to provide more information about the memory cost of the proposed method.

---

> ### Author Response · Authors · 2025-11-24
> **Rebuttal by Authors**
>
> We thank the reviewer for the comments.
>
>
> **Q1. BPTT.**
>
> The previous state is **explicitly detached** from the computation graph at each iteration. Concretely, at iteration $ t+1 $ we take the **hidden state** $ h^{t} $ produced at iteration  $ t $, treat it as a **fixed input feature**, and backpropagate the loss only through the computations of the current iteration (i.e., no backpropagation through time over the whole sequence $ \{h^{0},\dots,h^{t}\} $ ). This design choice is deliberate: (i) BPTT over tens of thousands of iterations on graphs with up to $ 10^6 $ nodes would be prohibitive in memory and runtime, and (ii) our goal is to learn a *local update operator* that improves the current solution given the previous one, rather than to optimize the full unrolled trajectory. Empirically, this “memoryless” iterative refinement is sufficient to obtain stable and high-quality solutions, as evidenced by the narrow result distributions and clear gains over the non-iterative baseline (see for example  Figures 3 with ablations). In short, we detach the previous state and do not use BPTT across iterations.
>
>
> **Q2. Stronger heuristics.**
>
> In Table 1 we followed the **exact BA-graph Max-Cut setup and baselines from [1]**, which reported only MFA and Greedy as the heuristics for this benchmark, but the table already includes **Gurobi**, which consistently dominates standard heuristic, Since QIGNN closely matched or even surpassed Gurobi on the larger BA graphs, we did not expect additional heuristic baselines  to change the qualitative conclusion. However, because of the request of another reviewer, we will add changes in this table, including adding additional stronger heuristics, please follow paper version updates.
>
> **Q3. G70 solution improvement**
>
> Firstly, It is important to note that, prior to our work, no neural method had approached the performance of BLS on Gset. In fact, most neural approaches consider BLS as a benchmark or ground truth (e.g., [2]).
>
> Secondly, Although the statistics for a single graph are insufficient to draw a generalizable conclusion, the behavior on G70 is consistent with what we observe on *other* benchmarks: in Tables 1 and 5, QIGNN is particularly strong on the largest graphs, where it matches or surpasses very strong solvers (e.g., Gurobi, KaMIS) while remaining much faster. Our intuition is that local-search heuristics like BLS/TSHEA perform sequential single variable updates, so for a fixed time budget the effective number of “useful updates per node” decreases as the graph grows.  In contrast, QIGNN performs parallel global updates via message passing and the iterative state, so each iteration refines all nodes at once, and this becomes increasingly beneficial on larger graphs. Appendix Fig. 4(a) already shows that QIGNN’s runtime scales essentially linearly with the number of nodes.
>
>
> **Q4.** The same intuition about larger instances applies to Tables 3 and 4 as well, and this is supported by runtime measurements: in Appendix Table 8, QIGNN solves the PUBMED instance with ~20,000 nodes in about 3 minutes.
>
> **Q5.** Please see Appendix Fig. 4 that shows that QIGNN’s runtime scales essentially linearly with the number of nodes for random d-regular graphs.
>
> **Q6.** Difusco and T2TCO are **supervised** solvers that require large datasets, and our statement about ER-9000–11000 follows [1]: these supervised architectures are *not scalable* to that regime and therefore omit them. In contrast, QIGNN is **instance-wise unsupervised** and much lighter in memory: we only store a single sparse graph and per-node hidden states.   This allows QIGNN to comfortably handle ER graphs with 9k–11k nodes on a single GPU, whereas the transformer/diffusion supervised baselines become memory and runtime constrained in this regime.
>
>
>
>
>
>
>
>
> [1] Let the Flows Tell: Solving Graph Combinatorial Optimization Problems with GFlowNets, neurips 2023
>
> [2] Controlling Continuous Relaxation for Combinatorial Optimization. NeurIPS 2024

---

> > ### Comment · Reviewer_BTQ4 · 2025-11-26
> >
> > On Q2:
> > Gurobi is not a strong reference solver for comparing against state-of-the-art Max-Cut heuristics. I look forward to seeing the updated results. Why not include EO, given that it is a strong heuristic and was already reimplemented according to the paper?
> >
> > On Q3:
> > The explanation about parallel versus sequential updates does not fully address concerns about search-dynamic quality. The appeal to consistency with Table 1 is also unconvincing unless Table 1 includes stronger heuristics (see Q2).
> >
> > In Table 5 specifically, does QIGNN actually achieve better MIS solutions than KaMIS at comparable runtime?
> > QIGNN is faster, but the MIS values appear substantially smaller.
> >
> > On Q5:
> > Figure 4 shows scaling for a fixed number of iterations, but the relevant metric is wall-clock time to reach a target solution quality. This was not addressed.

---

> > > ### Author Response · Authors · 2025-11-27
> > > **Official Comment by authors**
> > >
> > > We thank the reviewer for their response.
> > >
> > > **On Q2.** We have now applied our EO implementation to the BA Max-Cut graphs used in Table 1. The results are:
> > >
> > > | Method       | Type | BA-small Size ↑ | BA-large Size ↑ |
> > > |-------------|------|------------------|------------------|
> > > | EO          | H   | 729.51     | 2921.31     |
> > > | QIGNN  (from paper)     | UL   | 732.2     | 2965.85     |
> > >
> > > QIGNN demonstrates superior performance compared to EO on both small and large BA graphs. Results for EO will be included in the corresponding table in the revised version of the paper. We will update the text of the paper, including Table 1, till the end of rebuttal stage, and we will add a general summary of these revisions.
> > >
> > > **On Q3 / Table 5.** For the MIS problem on the large ER-9000–11000 graphs presented in Table 5, QIGNN outperforms KaMIS by obtaining better solutions at significantly lower runtime (please see the right column of Table 5). Specifically, the average independent set size for QIGNN is 375.44, compared to 374.57 for KaMIS, while QIGNN’s average runtime is approximately 40 times smaller. This result provides evidence that the global updates employed in our iterative refinement process lead to a higher probability of finding good minima on large graphs within realistic time constraints, even when compared to SOTA local-search solvers such as KaMIS.
> > >
> > > **On Q5.** We agree that time-to-target is the most relevant metric. While Fig. 4(a) focuses on runtime vs. problem size for a fixed number of iterations, Fig. 4(b) already provides a proxy for time-to-target: for graphs up to \(10^6\) nodes it shows at which iteration (epoch) QIGNN finds its best solution, and each iteration has approximately constant cost.

---

### Official Review · Reviewer_SZPE · 2025-10-31

**Soundness:** 4
**Presentation:** 2
**Contribution:** 3
**Rating:** 6
**Confidence:** 4

**Summary:**

This paper proposes QIGNN, a novel unsupervised Graph Neural Network (GNN) framework for solving Combinatorial Optimization (CO) problems. To address the issue of existing GNNs getting trapped in local optima by predicting solutions in a single pass from fixed node features, it introduces an iterative refinement mechanism that feeds the predictions from the previous iteration back as dynamic node features for the subsequent step. The authors claim their method significantly outperforms prior learning-based approaches on Max-Cut, Graph Coloring, and Maximum Independent Set (MIS) problems, and that on large-scale instances, it matches or surpasses state-of-the-art classical heuristics while being more computationally efficient.

**Strengths:**

1, The core idea of iterative refinement is not, in itself, a new concept within the optimization field. However, its specific application within an end-to-end, unsupervised GNN framework for QUBO problems—by feeding the entire output from a previous step back as a dynamic input feature for the next—demonstrates a reasonable degree of originality. Other architectural elements (e.g., the parallel SAGEConv layers in Section 3.2) are more akin to a clever combination of existing techniques. Overall, while not paradigm-shifting, the contribution is a practical idea that effectively enhances existing GNN-based CO solvers.

2, The proposed QIGNN consistently outperforms state-of-the-art learning-based methods (e.g., PI-GNN, RUN-CSP, various SL/RL methods) across a wide range of benchmarks on three distinct NP-hard problems: Max-Cut, Graph Coloring, and MIS (Tables 1, 2, 3, 4, 5, 6). This trend is particularly pronounced on large-scale graph instances.

3, The paper clearly demonstrates that the iterative refinement mechanism is the key driver of its performance gains. The ablation study (Figure 7), which shows a sharp performance degradation on the Max-Cut problem when this mechanism is removed, strongly supports the validity of the proposed design.

**Weaknesses:**

1, The authors state, "Since QIGNN is sensitive to initialization, we run multiple seeds and report the best result" (lines 268-269). Reporting the single best score from multiple runs, rather than the mean and standard deviation, can significantly inflate the method's perceived performance. This is a major flaw that severely harms reproducibility. It is unclear whether the baselines were subjected to the same "best-of-N" evaluation protocol (e.g., RUN-CSP is noted as best-of-64, line 314), casting doubt on the fairness of the comparison.

2, The paper uses different sets of baselines for the Max-Cut (Tables 1, 2), Graph Coloring (Tables 3, 4), and MIS (Tables 5, 6) problems. While including problem-specific SOTA heuristics is appropriate, several modern GNN-based methods are only compared on a subset of problems, making it difficult to form a holistic judgment of the proposed method's superiority.

3, While the authors acknowledge the method's sensitivity to initialization (line 268), they provide no analysis as to why this sensitivity arises or whether the iterative design might amplify it. This leaves open questions regarding the method's stability and practical reliability.

**Questions:**

1, Regarding the evaluation protocol of reporting the single best result from multiple runs (lines 268-269), I have a question about the statistical representation of the performance. For methods that are sensitive to initialization, a conventional approach is often to define a single 'evaluation trial' as the process of executing N runs and selecting the best outcome. The final reported metrics would then be the mean and standard deviation over several such independent trials to reflect the expected performance and its variance. Could you elaborate on the rationale for reporting a single best-of-N result rather than the statistics over multiple trials? This clarification would help in assessing the practical reliability and expected performance of the proposed method.

2, In Table 2, QIGNN underperforms BLS/TSHEA on smaller Gset instances but outperforms them on the largest instance, G70. What is the intuition behind this superior scaling behavior? Please provide an analysis of how the proposed iterative refinement mechanism interacts with graph properties like size or density to become more effective on larger-scale problems.

---

> ### Author Response · Authors · 2025-11-24
> **Rebuttal by Authors**
>
> We thank the reviewer for the comments.
>
> **W1, Q1.**
>
>
> We agree that evaluation protocols must be fair. When we report “best-of-\(k\)” for QIGNN, we do so only in settings where the *baselines are also evaluated with multiple restarts*. For example, on Gset and random regular graphs, RUN-CSP, EO, pi-gnn is run multiple times and reported by its best value, and classical heuristics such as BLS are also multi-start heuristics. We use the same number of runs for QIGNN and PI-GNN. At the same time, many of our experiments already report averages: e.g., for MIS  (Tables 5, 6), table 1 for max-cut. Additionally, For Gset, Figures 3/5/6/7 show the distribution of QIGNN scores over 20 seeds: the spread is small and the median is close to the maximum, indicating that QIGNN is not relying on rare “lucky” initializations.
>
> Also, when GPU memory allows, we can construct a single composite graph with duplicated instances  and process in parallel. Thus, best-of-\(k\) for QIGNN also corresponds to a *parallel* multi-start procedure rather than \(k\) fully sequential runs. (lines 721-724)
>
> **W2.** We agree that having a uniform set of baselines across all problems would be ideal, but in practice this is difficult: we run 7 experiments across 3 different problems and multiple benchmark families, and many baselines are tailored to specific tasks or graph regimes (and not always available as reusable code). Our design choice was therefore to compare, for each benchmark, against the **strongest classical and neural methods that are known** for that setting, rather than forcing the same (possibly weak) baselines everywhere.
>
> Regarding GNN-based methods, our method is **unsupervised instance-specific**, so we first focused on a fair comparison to other unsupervised / QUBO-based neural methods on the relevant benchmarks (Tables 2, 3, 4, 6, 7). We then additionally compared to **supervised and RL-based methods** (Tables 1 and 5) and observed that QIGNN not only outperforms other unsupervised approaches, but is almost always better than *all* SOTA neural baselines by a clear margin. Moreover, on large graphs we match or surpass the strongest classical solvers such as KaMIS and Gurobi.
>
> **W3.** QIGNN is heuristic and, like all methods of this kind, does not guarantee an optimal solution for NP-hard problems, meaning it may converge to different local minima. The primary objective of heuristic algorithms is to rapidly find solutions that are near-optimal. We show empirically that QIGNN reliably converges to *good* minima with strong scalability. In particular, on large instances QIGNN consistently reaches solutions that are competitive with or better than state-of-the-art heuristics.
>
> We use standard stochastic initialization: all network weights are initialized with PyTorch defaults, node features contain a random component plus deterministic structural features, and the iterative state is initialized to zero. As with essentially any neural method or classical algorithm that contain any stochastic part, different random seeds can lead to slightly different solution; this is what we meant by “sensitive to initialization”, not that the method is unstable in practice. In fact, we deliberately *do not* tune hyperparameters per instance or dataset: the same default architecture and hyperparameters are used across all problems and benchmarks, unlike some baselines that are retuned per setting.
>
> Figure 3 shows the distribution of results over 20 seeds on several Gset graphs, with and without the iterative refinement. The spread is small, and the whole distribution for QIGNN with refinement is consistently shifted upwards, indicating that the iterative design  *improves* the solution for almost every seed. Further ablations in Figures 5–9 demonstrate that QIGNN is also robust to changes in learning rate, architecture and etc. We will clarify line 268 in the revised paper.
>
>
> **Q2.**  Our GNN shows near-linear scalability (see Fig. 4a), and under runtime constraints, it has a higher probability of finding a good solution on large graphs compared to exact solvers and the heuristics considered. Local-search heuristics like BLS/TSHEA get fewer “useful updates per node” as graphs grow, while QIGNN’s message passing and iterative state update refine *all* nodes in parallel at each step. This means that on large graphs, giving QIGNN more iterations within a similar time budget leads to consistently larger gains in solution quality.
>
> Although the statistics for a single graph are insufficient to draw a generalizable conclusion, the behavior on G70 is consistent with what was observed across *other* benchmarks: in Tables 1,5 QIGNN is especially strong on the **larger** instances, where it matches or surpasses the other solvers while remaining much faster.
>
> We also provide experiments for graphs with different densities, (please see table 7 with d-regular graphs).

---

> > ### Comment · Reviewer_SZPE · 2025-11-28
> >
> > I thank the authors for the detailed response. The clarifications on the evaluation protocol (Best-of-N) and the scalability intuition (parallel updates vs. sequential local search) have effectively addressed my concerns.
> >
> > I decided to maintain my score of 6. While the proposed method is effective and the "iterative refinement" is a sensible design choice, the core technical novelty feels more evolutionary (a clever combination of existing techniques) rather than a fundamental breakthrough.

---

### Official Review · Reviewer_Twr2 · 2025-11-01

**Soundness:** 2
**Presentation:** 2
**Contribution:** 1
**Rating:** 2
**Confidence:** 4

**Summary:**

This paper introduces QIGNN, an unsupervised iterative optimization framework for solving COPs formulated as QUBO. Compared to PIGNN, the proposed method incorporates an iterative dynamic feedback mechanism that helps the model escape local minima. Experiments on classic benchmarks such as MaxCut, MIS, and Graph Coloring demonstrate that QIGNN achieves superior solution quality over several learning-based approaches, while maintaining high efficiency on large-scale graphs.

**Strengths:**

1. The authors identify the limitations of existing methods and propose an iterative dynamic feedback mechanism to address them.
2. Experiments on classic benchmarks demonstrate the effectiveness of QIGNN in terms of solution quality and computational efficiency.

**Weaknesses:**

1. Although the paper introduces a dynamic iterative mechanism to enhance performance, it lacks a clear analysis of the mechanism’s complexity in the embedding and feature spaces, which may lead to potential computational overhead.
2. The paper does not include sufficient competitive baselines to fully validate the performance of QIGNN. Moreover, since QIGNN is trained and tested on the same instances, the authors should report the overall training and testing complexity together for a fair evaluation.
3. While the paper targets general COPs, the experiments are limited to only three classic benchmarks, which restricts the demonstration of its general applicability.

**Questions:**

1. Could you include some competitive baselines to better illustrate the performance of QIGNN, and incorporate these methods into the related work section (e.g., [1–3])?
2. Could you train QIGNN on a set of instances and directly apply it to test instances drawn from the same distribution to evaluate its generalization ability?
3. Although COPs can theoretically be transformed into QUBO formulations, different COPs exhibit varying levels of difficulty during transformation. Could you present additional problems that can be converted into QUBO form, like routing, packing problems,  and apply QIGNN to them to further demonstrate its effectiveness? Moreover, please report the transformation complexity along with the training and testing complexities. In addition, transforming some COPs into QUBO formulations requires introducing auxiliary variables, which causes the dynamic mechanism to incur increasing computational overhead.
4. Could you provide a detailed analysis of the computational complexity of QIGNN and compare it with the baselines?
5. Could you provide a more detailed analysis of the hyperparameters? For instance, the penalty coefficient that governs the transformation from COPs to QUBO formulations directly influences the problem structure and solution quality.

[1] Learning to Solve Quadratic Unconstrained Binary Optimization in a Classification Way. NeurIPS 2024.
[2] Controlling Continuous Relaxation for Combinatorial Optimization. NeurIPS 2024.
[3] Scalable Discrete Diffusion Samplers: Combinatorial Optimization and Statistical Physics. ICLR 2025.

---

> ### Author Response · Authors · 2025-11-24
> **Rebuttal by Authors part 1.**
>
> We thank the reviewer for the comments.
>
> **W1.** We agree that the complexity of the iterative mechanism should be clearly stated. The key point is that our refinement step does **not** change the asymptotic cost of the model: each iteration has the same complexity as a standard GNN forward pass, and the only overhead is that we process a slightly larger node feature vector (we add one dynamic feature channel containing the node state  from the previous epoch on top of the static features). In other words, we go from $ x $ to $ x+1 $ feature dimensions per node; the number of message-passing layers and the overall scaling in $ |V| $ and $ |E| $ remain unchanged. Appendix Fig. 4 empirically confirms that QIGNN’s runtime scales essentially linearly with the number of nodes.
>
> **W2.**  First of all, QIGNN is an **unsupervised, instance-wise** method: it is **not** trained on a separate dataset of solved problems, but directly optimizes the objective of a **single given instance** in the end-to-end manner: each graph is optimized **independently**, without access to labeled solutions or a training dataset (similarly to running a heuristic like local search). In other words, there is **no train/test split** and the “training loop” *is* the solving procedure for that instance. We kindly ask the reviewer to revisit Sections 2 and 3, where this unsupervised setup is described, as it is a key conceptual point of the method.
>
> **W3.** We agree that demonstrating general applicability is important, but we believe our current experimental setup already goes  beyond what is standard in this area. Although we focus on three classic COP families (Max-Cut, MIS, and Graph Coloring with 2 formulations), we evaluate QIGNN on **seven** benchmark regimes, including both synthetic and real-world graphs, with very different structures and QUBO formulations (e.g., Gset vs. BA graphs for Max-Cut, two distinct Coloring setups, and several MIS benchmarks. These cover a broad range of graph sizes (up to >20k nodes), densities, and constraint structures, as well as baselines, and QIGNN uses the *same* architecture and hyperparameters across them, which we see as a strong indication of generality rather than overfitting to a single niche setting.
>
> **Q1.**
> We thank the reviewer for proposing additional baselines to compare with.
> Firsly, we already compare with [2] on MIS problem - please see CRA in table 5, and show that QIGNN significantly surpasses [2]. We have also added CRA [2] results for Gset graphs for MAX-CUT problem (please check revised version of the paper): QIGNN matches or significantly outperforms CRA on the all GSET graphs.
>
> Secondly, compared to our setting, [3] evaluated SDDS on different graph distributions. We have now run additional MIS experiments on RB graphs, and in the new table QIGNN outperforms the best SDDS results on both small and large instances:
>
> | Method       | Type | RB-small Size ↑ | RB-large Size ↑ |
> |-------------|------|------------------|------------------|
> | Gurobi      | OR   | 20.13 ± 0.03     | 42.51 ± 0.06     |
> | SDDS (best) | UL   | 19.62 ± 0.01     | 39.99 ± 0.08     |
> | QIGNN       | UL   | 20.07 ± 0.12     | 41.22 ± 0.37     |
>
> We will include the full table in the appendix. Note that although SDDS is unsupervised, it still requires training on a collected dataset of graphs from the same distribution it is later applied to, which limits its generalization and practical usability, whereas QIGNN does not require any training set and directly optimizes a single given instance.
>
> [1]  is evaluated on synthetic QUBO benchmarks and large dense QUBO test sets while our experiments are on QUBO formulations of 3 CO problems. Their architecture and training pipeline are tailored to arbitrary Q matrices rather than graph-structured inputs, and  do not report results on these tasks. Also their training procedure is not fully unsupervised in the sense of our method. They rely on a local search (BGF) and a historical buffer to construct pseudo-labels, and then train the network with a supervised loss against these pseudo-labels. This is closer to pseudo-labeling / self-supervised distillation to the purely unsupervised GNN setting we consider.
> Additionally, there is no publicly available implementation of their method so we cannot apply it to our problems and we already compare with a wide range of high-quality baselines.

---

> ### Author Response · Authors · 2025-11-24
> **Rebuttal by Authors part 2.**
>
> **Q2.** Our model is **unsupervised and instance-wise**: it optimizes a single problem instance in an end-to-end manner, i.e., each graph is optimized independently without access to labeled solutions or to a separate training set of instances. Therefore, there is no notion of a training set and test set drawn from a distribution. You can directly apply QIGNN to any single problem instance as heuristic optimization algorithm.
>
> **Q3.**
> We deliberately focused on Max-Cut, MIS, and Coloring because these are among the COP families with the richest **QUBO-based [2,4-6] and unsupervised CO** literature. We agree that not every problem is best attacked via a QUBO formulation, but QUBOs have proven to be one of the most effective and general modeling tools for a large class of CO problems, very much like MILP formulations. As with MILP, the QUBO transformation is a modeling step: once a 0–1 formulation is chosen, turning it into QUBO is mechanical (adding quadratic penalties), and its “complexity” is negligible compared to actually *solving* the resulting instance.
>
> As discussed above, QIGNN is an **unsupervised, instance-wise** solver, so there is **no separate training vs. testing complexity**: the relevant cost is simply the runtime of one QIGNN run on a given QUBO graph.
> We also note that it is not even well-defined relative to which original formulation one is counting these extra variables, since in practice auxiliary variables are an integral part of virtually any modern CO formulation; it does not make the approach less efficient or less general, but is simply part of expressing more complex problems within a unified framework.
>
>
> **Q4**
> Fig. 4 empirically reports how QIGNN’s wall-clock time grows with $ |V| $, showing near-linear scalability.
>
> QIGNN is built on a standard message-passing GNN, so its **theoretical complexity** is the same as other GNN-based QUBO solvers such as PI-GNN and CRA.  Importantly, QIGNN and PI-GNN/CRA have the *same* complexity, but PI-GNN requires expensive instance-specific hyperparameter tuning, effectively multiplying its runtime, whereas QIGNN uses a single hyperparameter configuration across all benchmarks, obtaining much better results.
>
> **Q5.**
> We provide a hyperparameter analysis for the learning rate in Fig. 5, where QIGNN shows stable performance. For the remaining hyperparameters, we deliberately use an almost **single configuration across all tasks and datasets** and still obtain highly competitive results; combined with the ablations in the appendix, this suggests that QIGNN is not overly sensitive to fine-grained tuning. Regarding the QUBO penalty coefficients, for Max-Cut and Coloring we follow standard choices from the QUBO literature, while for MIS we use a size-dependent penalty as described in the MIS section of the appendix. We do not further analyze it because designing QUBO encodings is a separate modeling problem and an active research topic on its own, whereas our goal  is to evaluate QIGNN given a reasonable QUBO formulation rather than to optimize QUBO penalties.
>
>
>
> [1] Learning to Solve Quadratic Unconstrained Binary Optimization in a Classification Way. NeurIPS 2024.
>
> [2] Controlling Continuous Relaxation for Combinatorial Optimization. NeurIPS 2024.
>
> [3] Scalable Discrete Diffusion Samplers: Combinatorial Optimization and Statistical Physics. ICLR 2025.
>
> [4] Combinatorial optimization with physics-inspired graph neural networks, 2022
>
> [5] Stefan Boettcher, Inability of a graph neural network heuristic to outperform greedy algorithms in solving combinatorial optimization problems like Max-Cut, 2022
>
> [6] Rethinking graph neural networks for the graph coloring problem , 2022.

---

### Comment · Area_Chair_FJTL · 2025-11-24
**Discussion Period**

Dear reviewers,

The discussion period is now open. Please use the “Official Comments” to engage in discussions about each other's reviews and the authors' rebuttal, and update your assessments or comments as appropriate.

Did the authors' rebuttal adequately address your concerns? We kindly ask that you update your reviews based on these discussions and your evaluation of the rebuttal, even if your overall assessment remains unchanged.

Thank you all for your contributions.

Best regards, AC

---

### Meta-Review · Area_Chair_mNDg · 2026-01-10

**Summary:**

There are several concerns raised by the reviewers. One common concern is that there is no discussion on the computational cost. Also,  there are common questions about the implementation details and experimental setup. The authors should compare their method to the strong baselines suggested by one reviewer. There are concerns that the proposed method is evaluated on three problem types (Max-Cut, MIS, Graph Coloring) and might not generalize to more complex problems like packing or routing. The method is sensitive to initialization and needs multiple rounds to pick the best result.

**Reviewer Concerns:**

The authors have provided a rebuttal that addresses multiple concerns from the reviewers. The authors clarify that QIGNN is instance-wise, so there is no training on a dataset, and it is the actual solver for a single graph instance. The evaluation is fair because "Best-of-N" is standard for comparison against baselines that are multi-start heuristics. The authors also provide new experiments against SDDS and show that QIGNN performs better. Regarding the computational cost, one issue is that the presented approach is not very fast compared to baselines. Another outstanding issue that the proposed method has not fully surpassed Gurobi yet. Regarding the technical novelty, a concern is a lack of fundamental breakthrough, with reviewers viewing the iterative refinement as a "clever combination of existing techniques" rather than a breakthrough. Also, it is still not clear if the method can work well on other complex CO problems. Given these limitations, the AC is leaning to reject the paper.

**Reviewer Scores:**

The reviewers gave scores of 2,4,6, and 6. Some reveiwers may change their scores as some concerns are clarified.

---

### Decision · Program_Chairs · 2026-01-26

Reject